# SCOUT: Spatial-Aware Continual Scene Understanding and Switch Policy for Embodied Mobile Manipulation

## Abstract

Coordinating navigation and manipulation with robust performance is essential for embodied AI in complex indoor environments. To address this, **SCOUT** (**S**patial-Aware **CO**ntinual Scene **U**nderstanding and Swi**T**ch Policy for Embodied Mobile Manipulation) is proposed, consisting of: 1) Spatial-Aware Continual Scene Understanding with a Scene Modeling Module for effective scene modeling and a Mask Query Module for precise interaction mask generation; and 2) Switch Policy that dynamically transitions between long-term navigation and short-term reactive planning when viable manipulation opportunities are detected. SCOUT achieves state-of-the-art performance on ALFRED benchmark, reaching 65.09% and 60.79% success rates in test seen and unseen environments respectively with step-by-step instructions, while maintaining consistently robust performance (61.24% / 56.04%) without detailed guidance for long-horizon tasks.

## 1 Introduction

The vision of autonomous robots capable of assisting humans in everyday household tasks has long been a driving force in robotics research (Levine et al., 2018; Pashevich et al., 2021; Min et al., 2022; Brohan et al., 2022). Recent advances in Embodied AI (Prabhudesai et al., 2020; Stepputtis et al., 2020; Song et al., 2023; Chi et al., 2023; Kim et al., 2024) have demonstrated remarkable progress in enabling artificial agents to perceive, navigate, and interact with their surroundings. Within this context, Embodied Mobile Manipulation (Shridhar et al., 2020; Yenamandra et al., 2023) emerges as a crucial next step, requiring agents to coordinate both navigation and sophisticated object manipulation while maintaining robust performance in dynamic, complex environments.

Embodied Mobile Manipulation poses key challenges for agents. Agents must understand complex dynamic 3D environments through limited egocentric views, requiring robust spatial reasoning. The task also demands sophisticated long-horizon planning to coordinate navigation and manipulation actions while maintaining goal-oriented behavior. Real-time adaptation to environmental feedback adds significant complexity. ALFRED (Shridhar et al., 2020) serves as a standard benchmark for Embodied Mobile Manipulation, requiring agents to follow natural language instructions to complete long-horizon household tasks involving multiple sub-tasks and object interactions.

As shown in Fig. 1, existing Embodied Mobile Manipulation methods suffer from three key limitations. First, **historical information loss** occurs in end-to-end methods such as Seq2Seq (Shridhar et al., 2020), MOCA (Singh et al., 2021), and E.T. (Pashevich et al., 2021) that directly map current observations and instructions to actions without persistent spatial memory, causing agents to repeatedly search for previously encountered objects and reducing execution efficiency. Second, **inconsistent scene representation** affects modular methods like FILM (Min et al., 2022), CAPEAM (Kim et al., 2023), and DISCO (Xu et al., 2024) that use separate models for depth prediction, semantic segmentation, and affordance estimation before lifting semantic information to 3D using predicted depth. This pipeline causes error accumulation from inaccurate depth-semantic lifting and limits spatial reasoning due to restricted 2D feature interaction. Third, **rigid execution strategies** characterize methods (FILM (Min et al., 2022), Prompter (Inoue & Ohashi, 2022), and CAPEAM (Kim et al., 2023)) that rely heavily on hand-crafted strategies for scenarios like object state changes and closer target appearances, limiting adaptability and generalization.

Figure 1: Existing methods suffer from three key limitations: historical information loss, inconsistent scene representation, and rigid execution strategy. SCOUT addresses these issues through a scene representation that captures temporal consistency and spatial-semantic relationships, and a policy mechanism that flexibly switches between navigation and manipulation behaviors.

To address these key issues, we propose SCOUT, a novel framework with two components that is depicted in Fig. 1. **1) Spatial-Aware Continual Scene Understanding:** This component comprises two modules. The Scene Modeling Module uses cross attention that operates directly in 3D space to extract information from both current observation and historical feature. This design enables effective 3D feature interactions and updates scene feature to maintain spatial-semantic relationships and temporal consistency. Notably, SCOUT utilizes precise 3D-to-2D projection when extracting features from current observation, eliminating depth estimation dependence and improving accuracy. This enhances spatial reasoning capabilities for complex scene understanding. The Mask Query Module leverages 2D-3D feature alignment to generate interaction masks for object manipulation and enhances semantic richness of scene features. **2) Switch Policy:** A dual-planning approach where the long-term planner generates navigation trajectories while the short-term planner monitors for immediate interaction opportunities, enabling adaptive switching when viable interactions are detected. For fair evaluation on ALFRED (Shridhar et al., 2020), we adopt Min et al. (2022)'s language parsing module while focusing on advancing navigation and manipulation abilities.

In summary, our main contributions are three-folds:

- We propose a novel Spatial-Aware Continual Scene Understanding method with a Scene Modeling Module using cross attention in 3D space for effective scene modeling and a Mask Query Module utilizing 2D-3D feature alignment for interaction mask generation.

- We introduce a flexible Switch Policy enabling transitions between long-term and short-term planning, greatly improving success rate and efficiency of mobile manipulation tasks.

- Our approach achieves superior performance on ALFRED benchmark. Detailed ablation studies and thorough visualizations validate the effectiveness of each component.

## 2 RELATED WORK

**Embodied Mobile Manipulation.** Enabling mobile robots to navigate and manipulate objects in complex environments remains a major embodied AI challenge. Various simulators and benchmarks have emerged to advance this field (Kolve et al., 2017; Batra et al., 2020; Srivastava et al., 2021; Szot et al., 2021; Ehsani et al., 2023). While early works focused on navigation in static scenes, recent benchmarks require agents to manipulate objects in dynamic environments with evolving semantics. Some support physical robot testing (Yenamandra et al., 2023; Jaafar et al., 2024), while most use simulated environments for reproducible evaluation (Cheng et al., 2025; Yang et al., 2025). ALFRED (Shridhar et al., 2020) stands out by integrating navigation and manipulation, challenging agents to follow natural language instructions for long-horizon household tasks. Its dynamic environment with changing object states provides an ideal testbed for our proposed SCOUT framework, which coordinates spatial-semantic understanding with adaptive navigation-manipulation policies.

**3D Scene Understanding.** 3D scene understanding lets machines parse spatial-semantic properties of scenes from images and point clouds. Recent advances include NeRF (Mildenhall et al., 2021; Liu et al., 2023; Zhu et al., 2023), which uses neural networks for high-quality 3D reconstruction from 2D images, and 3D Gaussian Splatting (3DGS) (Kerbl et al., 2023; Yu et al., 2024; Chen & Wang, 2024), which enables real-time rendering in dynamic scenes. Bird's-Eye-View (BEV) views have proved effective for autonomous driving and robotics (Huang et al., 2021; Can et al., 2021; Philion & Fidler, 2020; Liang et al., 2022; Wang et al., 2021; Liu et al., 2022), giving a top-down view for spatial perception. Recent methods include perspective-to-BEV translation using MLPs (Pan et al., 2020) and spatio-temporal transformers for merging multi-view features (Li et al., 2022). These methods allow rich spatial-semantic fusion, making BEV ideal for scene models. We use BEV to build a spatio-temporally coherent model for embodied mobile manipulation.

**Neural Policy.** Neural policies use neural networks to map states to actions, essential for embodied AI by enabling adaptability in complex environments(Zhao et al., 2024; Min et al., 2022; Wu et al., 2024). Min et al. (2022) use neural networks to generate actions from semantic maps, but map-based methods lack precision for object grounding. Xu et al. (2024) address this with fine-action networks, improving alignment and reducing ambiguity. Motivated by neural policy's adaptability, we employ it for short-term planning, enabling reactive decisions from real-time observations.

# 3 METHOD

In this work, we present our proposed SCOUT. The overall architecture is shown in Fig. 2. We first introduce our Spatial-Aware Continual Scene Understanding framework (Sec. 3.1) that captures spatial-semantic relationships and maintains temporal consistency. We then propose our Switch Policy (Sec. 3.2) that coordinates navigation and manipulation behaviors. Finally, we show SCOUT's application on the ALFRED benchmark (Sec. 3.3) (Shridhar et al., 2020).

## 3.1 SPATIAL-AWARE CONTINUAL SCENE UNDERSTANDING

For embodied mobile manipulation tasks, an effective scene representation must capture spatial relationships (such as relative object positions) and semantic information of the scene, while maintaining temporal consistency across dynamic environments. Inspired by recent BEV scene representations (Li et al., 2022; Huang et al., 2021) that employ spatio-temporal transformers to unify multi-view features into BEV space, we propose a Spatial-Aware Continual Scene Understanding framework. Unlike original BEV methods using multi-view images, our framework autoregressively aggregates temporal images from a single camera to achieve multi-camera spatial coverage.

Specifically, our framework comprises two key components: a Scene Modeling Module that leverages 3D map semantic segmentation task to learn scene representations capturing spatial, semantic information while maintaining temporal consistency, and a Mask Query Module that leverages 2D image semantic segmentation task to enhance the semantic richness of scene features and generate interaction masks for robotic manipulation.

### 3.1.1 SCENE MODELING MODULE

To achieve accurate and efficient scene modeling, we introduce a spatial cross attention (SCA) (Li et al., 2022) mechanism to update scene feature from current observation. Additionally, to integrate historical scene feature and acquire spatial reasoning ability in 3D space, we propose a temporal cross attention (TCA) mechanism. Finally, we employ a 3D map semantic segmentation head to convert scene feature into semantic scene map for training.

**Spatial Cross Attention (SCA).** As shown in Fig. 3 (a) and (b), from a bird's-eye perspective, we partition the entire room into an $H \times W$ grid, where each grid cell is initialized with a query $\mathbf{Q}_p \in \mathbb{R}^{1 \times C}$ located at $p = (x, y)$ with $C$ dimension. These queries are then lifted into pillar-like queries, where each 2D grid $p = (x, y)$ is transformed into $\mathbf{N}_{\text{ref}}$ 3D reference points $(x, y, z_i)$ sampled along the height dimension. For each reference point, we project it onto the current observation through the projection matrix of agent's camera, which can be written as:

$$z_i' \cdot [x_i' \quad y_i' \quad 1]^T = \mathbf{P} \cdot [x \quad y \quad z_i \quad 1]^T . \tag{1}$$

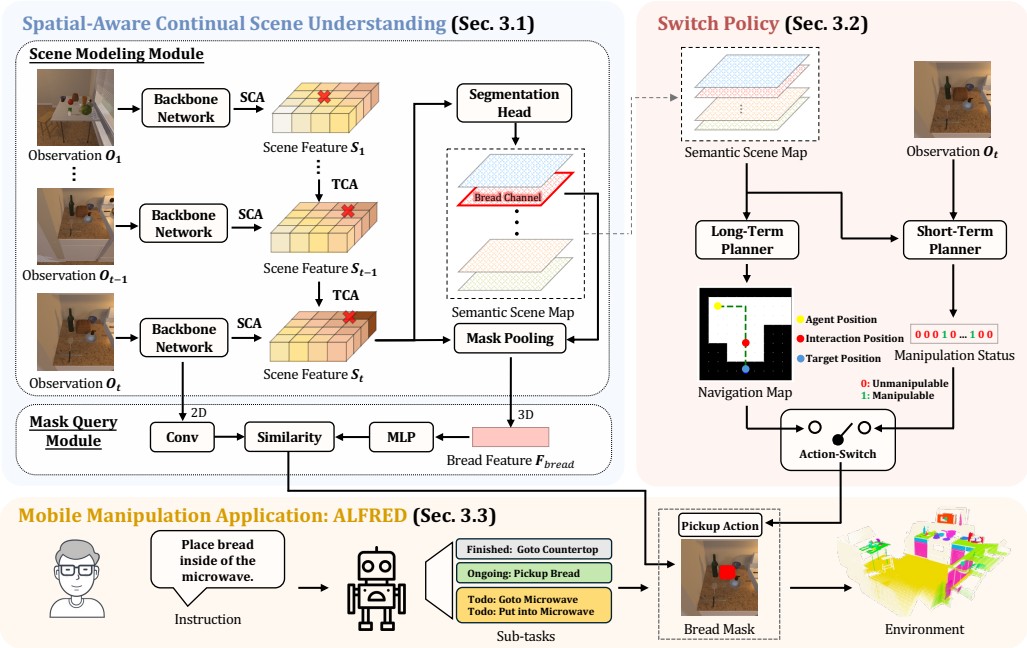

Figure 2: Our SCOUT method has two main parts: Spatial-Aware Continual Scene Understanding that handles observations through cross attention and feature alignment mechanisms to generate semantic scene maps and interaction masks, and Switch Policy that alternates between long-term navigation and short-term manipulation planning. The application shows placing bread inside a microwave, where the agent (marked by ✗) navigates and manipulates objects based on instructions.

Here, $(x_i', y_i')$ is the 2D image plane coordinate projected from 3D point $(x, y, z_i)$, $\mathbf{P} \in \mathbb{R}^{3 \times 4}$ is the known projection matrix of agent's camera, and $z_i'$ represents the depth factor of the projected point in the camera coordinate system. This projection yields a 2D reference point in the image plane, around which we sample and aggregate image features through weighted summation to obtain the output of SCA, formulated as follows:

$$\text{SCA}(\mathbf{Q}_p, \mathbf{F}_t) = \sum_{i=1}^{\mathbf{N}_{\text{ref}}} \text{DeformAttn}(\mathbf{Q}_p, (x_i', y_i'), \mathbf{F}_t), \quad (2)$$

where $i$ indexes the reference points and $\mathbf{F}_t$ is the multi-level semantic features extracted from the current frame $\mathbf{O}_t$ using a ResNet50 backbone (He et al., 2016). After applying this computation process to all queries in the scene, we finish updating the scene feature using the current observation.

Here, two points require clarification. First, vanilla multi-head attention (Vaswani et al., 2017) in our SCA is computationally expensive. Following Li et al. (2022), we adopt deformable attention (DeformAttn in equation 2) (Zhu et al., 2021), where each grid query $\mathbf{Q}_p$ only interacts with its regions of interest in the observation. Second, methods like FILM (Min et al., 2022) and DISCO (Xu et al., 2024) project 2D image features to 3D space using depth estimation, causing error accumulation. Our SCA instead projects 3D points to the image plane to extract 2D features for 3D scene updates, eliminating depth estimation dependence and improving modeling accuracy.

**Temporal Cross Attention (TCA).** As shown in Fig. 3 (b) and (c), given the scene query $\mathbf{Q}$ at current timestep $t$ and the cached scene feature $\mathbf{S}_{t-1}$ from timestep $t - 1$, each grid query $\mathbf{Q}_p$ at position $p = (x, y)$ fuses relevant features from $\mathbf{S}_{t-1}$ to achieve temporal consistency while interacting with neighboring features in $\mathbf{Q}$ to enable spatial reasoning. For computational efficiency, we adopt deformable attention to implement TCA, similar to SCA, which can be formulated as:

$$\text{TCA}(\mathbf{Q}_p, \{\mathbf{Q}, \mathbf{S}_{t-1}\}) = \sum_{\mathbf{V} \in \{\mathbf{Q}, \mathbf{S}_{t-1}\}} \text{DeformAttn}(\mathbf{Q}_p, p, \mathbf{V}). \quad (3)$$

Our TCA leverages cross attention to adaptively sample and aggregate scene features from the previous timestep, enabling scene memory and enhancing modeling consistency. Meanwhile, each

grid query interacts with surrounding scene features in 3D space, which better captures geometric relationships and enhances spatial reasoning such as relative positions and distances.

**3D Map Semantic Segmentation Head.** As shown in Fig. 2, our Scene Modeling Module processes current observation via backbone to generate image features, then uses SCA to extract features for scene updates while TCA fuses temporal and 3D spatial information, producing scene feature $\mathbf{S}_t \in \mathbb{R}^{H \times W \times C}$, where $H \times W$ denotes grid cells and $C$ represents feature dimension. Based on the scene feature, we introduce a 3D map semantic segmentation head to predict object category distribution on the grid map $\mathbf{D}_t$, trained with ground-truth labels $\mathbf{L}_t$ through:

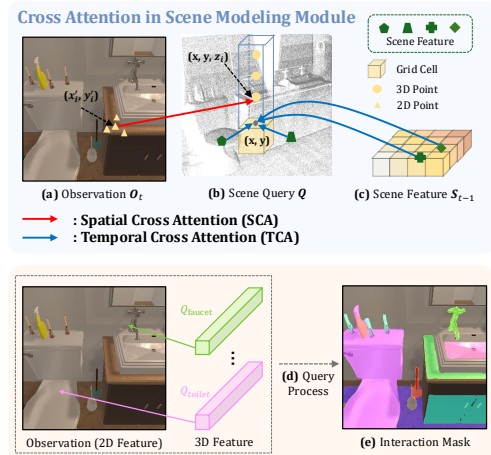

$$\mathcal{L}_{\text{map}} = \frac{1}{HW} \sum_{h=1}^{H} \sum_{w=1}^{W} \text{BCE}(\mathbf{D}_t^{h,w}, \mathbf{L}_t^{h,w}), \quad (4)$$

where $\mathcal{L}_{\text{map}}$ is the map segmentation loss, $\mathbf{L}_t$ is a multi-hot vector for each grid cell indicating the object categories present in that cell and BCE denotes binary cross-entropy loss measuring grid-wise discrepancy between predicted probabilities and ground-truth labels per category.

Figure 3: Demonstration of Cross Attention in Scene Modeling Module and Feature Alignment in Mask Query Module.

### 3.1.2 Mask Query Module

In embodied mobile manipulation tasks, accurate identification and localization of target objects are crucial for successful manipulation, typically achieved through semantic mask generation. Our Scene Modeling Module lacks this functionality, requiring us to develop methods to extract masks by leveraging the rich object features within our scene representation. Inspired by recent works (Peng et al., 2023; Schult et al., 2023) on 2D-3D feature alignment, we propose to generate target object mask by querying observation with object-specific feature from scene feature.

As shown in Fig. 2, for the **Pickup Bread** sub-task, we perform mask pooling over scene feature $\mathbf{S}_t$ using the bread channel from the semantic scene map. We aggregate features from bread-containing grid cells via averaging to compute $\mathbf{F}_{bread}$, then apply a multi-layer perception (MLP) to generate bread query $\mathbf{Q}_{bread}$. The general formula for computing object queries is:

$$\mathbf{F}_{object} = \frac{1}{|\mathcal{M}_{object}|} \sum_{(i,j) \in \mathcal{M}_{object}} \mathbf{S}_t^{(i,j)},$$

$$\mathbf{Q}_{object} = \text{MLP}(\mathbf{F}_{object}), \quad (5)$$

where $\mathcal{M}_{object}$ denotes grid cells where the object channel equals 1. Since object queries are derived from scene feature, they encode geometric properties and are termed 3D object features.

Simultaneously, convolutional layers are used to map backbone-extracted image features to new 2D object features. The similarity computation between 3D object queries and 2D object features produces object masks, precisely localizing objects for manipulation. The toilet scene in Fig. 3 (d) and (e) further demonstrates this 2D-3D feature alignment mechanism. To train Mask Query Module, we introduce a 2D image semantic segmentation loss $\mathcal{L}_{\text{img}}$ as follows:

$$\mathcal{L}_{\text{img}} = \frac{1}{HW} \sum_{h=1}^{H} \sum_{w=1}^{W} \text{BCE}(\mathbf{M}_t^{h,w}, \mathbf{G}_t^{h,w}), \quad (6)$$

where $\mathbf{M}_t^{h,w}$ represents the predicted image mask at pixel $(h, w)$ at time-step $t$, and $\mathbf{G}_t^{h,w}$ denotes the corresponding ground-truth segmentation label.

### 3.1.3 TRAINING DETAILS

**Overall Objective.** To enable scene feature to capture spatial relationships and semantic richness while maintaining temporal consistency, we propose joint supervision combining 3D map segmentation and 2D image segmentation losses. The total loss $\mathcal{L}$ is formulated as follows:

$$\mathcal{L} = \mathcal{L}_{\text{map}} + \lambda \mathcal{L}_{\text{img}}, \tag{7}$$

where $\lambda$ is a hyperparameter that balances the map and image segmentation losses.

**Implementation.** Following DISCO (Xu et al., 2024), we collect training instructions, images, semantic map and image labels from ALFRED (Shridhar et al., 2020) expert trajectories. For temporal continuity, semantic map labels at timestep $t$ incorporate object distributions from step 1 to current timestep. We use a ResNet50 (He et al., 2016) backbone, SCA and TCA layers with deformable attention, a map segmentation head, and a mask query module. On ALFRED, each scene is modeled as a $25m \times 25m$ room with $25cm \times 25cm$ grids, resulting in scene queries of size $H \times W \times C$ where $H = W = 100$ and $C = 256$. These scene queries serve as learnable parameters that are jointly optimized during training. The model is trained using AdamW with BCE loss for 25 epochs, batch size 64 and learning rate 2e-4. Training completes in 24 hours on 8 RTX 4090 GPUs.

## 3.2 SWITCH POLICY

In embodied mobile manipulation tasks, agents must navigate toward targets while monitoring for environmental changes. As shown in Fig. 4 (a), during a **Pickup Potato** sub-task, after detecting a potato and planning a route, the agent might discover another potato in a previous blind spot after turning right. Hence, we need a mechanism to interrupt the original navigation plan to immediately execute the pickup action on this closer target, enhancing efficiency. For this purpose, we propose Switch Policy, which combines a long-term planner and a short-term planner. Notably, at the beginning of each task, agents perform random exploration of the scene to discover target objects.

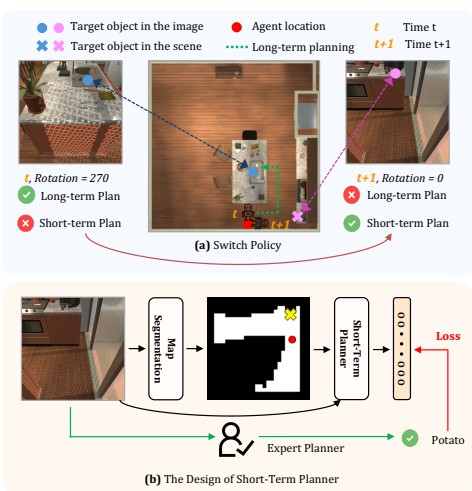

(a) Switch Policy

(b) The Design of Short-Term Planner

Figure 4: Demonstration of Switch Policy and our design of short-term planner.

**Random Exploration.** Before potato detection, the agent explores based on a navigable map. In the semantic scene map, a grid is considered **navigable** if only the floor category has a value of 1 while all other object categories are 0, thereby generating the navigable map. The agent randomly selects destinations from navigable cells and plans paths using BFS (Breadth-First Search).

**Long-Term Planner.** After potato detection, the agent switches to goal navigation. It locates the potato at the highest probability grid, expands with navigable grids within 1-meter radius, and uses BFS to generate navigation waypoints (grid coordinates) considering potato region, agent position, and navigable area. Thus, the long-term planner is a rule-based planner requiring no training.

**Short-Term Planner.** Due to the agent's limited field of view, the long-term planner may overlook information in occluded areas, causing unnecessary steps. To address this, we introduce a short-term planner that continuously monitors the environment for manipulation opportunities while executing the long-term plan. As shown in Fig. 4 (a), when discovering a potato within range during navigation, the short-term planner switches to pickup action, bypassing the remainder of the navigation path. We term this switching mechanism as Switch Policy, which improves efficiency by enabling direct manipulation with newly detected targets.

As shown in Fig. 4 (b), our short-term planner takes current observation and semantic scene map as inputs, outputting a binary vector indicating whether each object category is manipulable (1=manipulable, 0=unmanipulable). An object is manipulable only if semantically manipulable and within operational range, requiring precise understanding of semantic information in observation and spatial relationships in the semantic scene map. To train the short-term planner, we design an expert

planner with full scene knowledge for data collection. Following ALFRED (Shridhar et al., 2020) benchmark data generation process, our expert planner annotates manipulable target categories for each frame, yielding 332,432 training images.

The short-term planner employs ResNet50 (He et al., 2016) as backbone for visual feature and map feature extraction, followed by a linear classifier predicting manipulable object categories. The model is trained using BCE loss based on manipulation labels from the expert planner. Training utilizes AdamW optimizer with learning rate of $5 \times 10^{-5}$ and batch size of 128 for 40 epochs to achieve optimal performance. Training completes in 2 hours on 8 RTX 4090 GPUs.

Our Switch Policy systematically generates actions by leveraging both planners' advantages. For multi-instance targets (potatoes, watches, apples), the short-term planner discovers new goals to improve efficiency. For unique large targets (fridges, sidetables, sofas), the semantic scene map provides precise memory, making long-term routes highly accurate. When the short-term planner fails due to occlusion or inappropriate pose, our hierarchical fallback strategy continues executing existing long-horizon plans or generates new navigation plans to reachable targets.

### 3.3 MOBILE MANIPULATION APPLICATION: ALFRED

We evaluate our approach on ALFRED (Shridhar et al., 2020), a widely-adopted benchmark for embodied instruction following (specifications in Appendix A). Each ALFRED task contains high-level goal descriptions and step-by-step instructions. Appendix A.5 shows an ALFRED task **Put a cooked potato slice on the counter** where our agent only needs to place any cooked potato slice on any counter to complete the task. This allows our approach to focus on semantic segmentation to identify object categories rather than distinguishing individual instances.

Fig. 2 illustrates the SCOUT pipeline on ALFRED. Based on received instruction, the agent converts it into executable sub-tasks. For fair comparison with baselines (Min et al., 2022; Kim et al., 2023; Xu et al., 2024), we adopt Min et al. (2022)'s instruction processing module. For each sub-task, the agent employs Spatial-Aware Continual Scene Understanding to perceive the environment and generate accurate semantic scene map and interaction mask, while employing Switch Policy to alternate between long-term and short-term planning for improved efficiency. Details are in Appendix B.

## 4 EXPERIMENTS

### 4.1 DATASET AND METRICS

We conduct experiments on ALFRED (Shridhar et al., 2020), a benchmark dataset for vision-language navigation and long-horizon interaction tasks. The dataset comprises 25,726 episodes: 21,023 for training, 1,641 for validation, and 3,062 for testing. Both validation and test sets are divided into seen and unseen environments, with validation containing 820/821 episodes and test containing 1,533/1,529 episodes for seen/unseen scenes. We evaluate our method against competitive baselines using the test split, while using the validation split for analytical studies.

We employ four metrics: Success Rate (**SR**) measures completed task proportion; Goal Condition (**GC**) measures the percentage of met goal conditions for partial task success; Path Length Weighted Success Rate (**PLWSR**) and Goal Condition (**PLWGC**) incorporate trajectory efficiency by comparing agent paths with expert demonstrations. SR and GC assess task effectiveness, while PLW variants assess execution efficiency. Higher values indicate better performance across all metrics.

### 4.2 COMPARISON WITH THE STATE OF THE ART

We evaluate our method against competitive baselines on ALFRED benchmark. These include Seq2Seq (Shridhar et al., 2020), MOCA (Singh et al., 2021), E.T. (Pashevich et al., 2021), HLSM (Blukis et al., 2022), ABP (Kim et al., 2021), FILM (Min et al., 2022), LGS-RPA (Murray & Cakmak, 2022), Prompter (Inoue & Ohashi, 2022), CAPEAM (Kim et al., 2023) and DISCO (Xu et al., 2024). All these methods leverage RGB visual inputs and language instructions during inference. For fair comparison, we present results separately based on whether step-by-step instructions are utilized. By default, our SCOUT operates with high-level goal descriptions, though it can also incorporate step-by-step instructions when available.

Table 1: Performance comparison on ALFRED benchmark (Shridhar et al., 2020). ✓/ ✗ denotes whether step-by-step instructions are used. Bold values indicate the best results for each metric.

| | Ins | Test Seen | | | | Test Unseen | | | |
|---|---|---|---|---|---|---|---|---|---|
| | | SR | GC | PLWSR | PLWGC | SR | GC | PLWSR | PLWGC |
| Seq2Seq (Shridhar et al., 2020) | ✓ | 4.00 | 9.40 | 2.00 | 6.30 | 0.40 | 7.00 | 0.10 | 4.30 |
| MOCA (Singh et al., 2021) | ✓ | 26.81 | 33.20 | 19.52 | 26.33 | 7.65 | 15.73 | 4.21 | 11.24 |
| E.T. (Pashevich et al., 2021) | ✓ | 38.42 | 45.44 | 27.78 | 34.93 | 8.57 | 18.56 | 4.21 | 11.46 |
| ABP (Kim et al., 2021) | ✓ | 44.55 | 51.13 | 3.88 | 4.92 | 15.43 | 24.76 | 1.08 | 2.22 |
| FILM (Min et al., 2022) | ✓ | 27.67 | 38.51 | 11.23 | 15.06 | 26.49 | 36.37 | 10.55 | 14.30 |
| LGS-RPA (Murray & Cakmak, 2022) | ✓ | 40.05 | 48.66 | 21.28 | 28.97 | 35.41 | 45.24 | 15.68 | 22.76 |
| Prompter (Inoue & Ohashi, 2022) | ✓ | 51.17 | 60.22 | 25.12 | 30.21 | 45.32 | 56.57 | 20.79 | 25.80 |
| CAPEAM (Kim et al., 2023) | ✓ | 51.79 | 60.50 | 21.60 | 25.88 | 46.11 | 57.33 | 19.45 | 24.06 |
| DISCO (Xu et al., 2024) | ✓ | 59.50 | 66.10 | 40.60 | 47.40 | 56.50 | 66.80 | 36.50 | 44.50 |
| **SCOUT (Ours)** | ✓ | **65.09** | **72.88** | **52.42** | **58.29** | **60.79** | **68.60** | **46.82** | **52.57** |
| HLSM (Blukis et al., 2022) | ✗ | 25.11 | 35.79 | 6.69 | 11.53 | 16.29 | 27.24 | 4.34 | 8.45 |
| FILM (Min et al., 2022) | ✗ | 25.77 | 36.15 | 10.39 | 14.17 | 24.46 | 34.75 | 9.67 | 13.13 |
| LGS-RPA (Murray & Cakmak, 2022) | ✗ | 33.01 | 41.71 | 16.65 | 24.49 | 27.80 | 38.55 | 12.92 | 20.01 |
| Prompter (Inoue & Ohashi, 2022) | ✗ | 47.95 | 56.98 | 23.29 | 28.42 | 41.53 | 53.69 | 18.84 | 24.20 |
| CAPEAM (Kim et al., 2023) | ✗ | 47.36 | 54.38 | 19.03 | 23.78 | 43.69 | 54.66 | 17.64 | 22.76 |
| DISCO (Xu et al., 2024) | ✗ | 58.00 | 64.90 | 39.60 | 46.50 | 54.70 | **65.50** | 35.50 | 43.60 |
| **SCOUT (Ours)** | ✗ | **61.24** | **68.74** | **46.89** | **53.22** | **56.04** | 64.79 | **41.82** | **48.04** |

Table 2: Ablation study results. w/ GT masks denotes using ground-truth semantic segmentation masks from simulator, w/ SQ denotes using static query for interaction mask generation, and w/ Grid 80/120 denotes scene grid resolutions of 80×80 and 120×120.

| | Valid Seen | | | | Valid Unseen | | | |
|---|---|---|---|---|---|---|---|---|
| | SR | GC | PLWSR | PLWGC | SR | GC | PLWSR | PLWGC |
| SCOUT | 62.32 | 66.57 | 44.24 | 49.39 | 61.27 | 69.01 | 38.06 | 42.63 |
| w/ GT masks | 62.56 | 67.52 | 44.66 | 49.93 | 63.82 | 72.31 | 39.69 | 44.30 |
| w/ SQ | 60.98 | 65.15 | 42.38 | 47.04 | 58.83 | 66.18 | 36.18 | 40.37 |
| w/ Grid 80 | 52.68 | 60.08 | 40.62 | 46.22 | 52.25 | 60.94 | 30.40 | 34.47 |
| w/ Grid 120 | 53.54 | 59.60 | 38.98 | 44.65 | 52.86 | 62.40 | 33.94 | 39.18 |
| w/o SCA | 53.54 | 62.30 | 38.48 | 48.03 | 52.74 | 65.47 | 35.70 | 40.51 |
| w/o TCA | 42.80 | 53.53 | 19.56 | 30.51 | 27.16 | 40.42 | 9.51 | 18.43 |
| w/o Switch Policy | 57.59 | 62.74 | 38.44 | 43.63 | 57.13 | 63.39 | 34.57 | 38.07 |

We present comprehensive comparisons with state-of-the-art methods on ALFRED benchmark in Table 1. Our SCOUT framework demonstrates superior performance across most evaluation metrics in both instruction settings. With step-by-step instructions, SCOUT achieves 65.09% and 60.79% success rates on seen and unseen test splits respectively, surpassing the previous best method DISCO by 5.59% and 4.29%. Without instructions, SCOUT maintains strong performance with 61.24% and 56.04% success rates, representing improvements of 3.24% and 1.34% over DISCO.

Notably, SCOUT exhibits substantial improvements in path-length-weighted metrics, achieving 52.42% PLWSR and 58.29% PLWGC on seen environments, and 46.82% PLWSR and 52.57% PLWGC on unseen environments. These significant gains validate the effectiveness of our switch policy in enhancing task completion efficiency. The consistent superior performance across metrics and instruction settings shows SCOUT's robustness and strong generalization capability, indicating strong potential for real-world applications where detailed instructions may not be available.

## 4.3 ABLATION STUDY

We conduct ablation studies on ALFRED validation splits to evaluate each component in SCOUT (Table 2). Results are averaged over 3 runs with different random seeds. Detailed experimental configurations for each ablation setting can be found in Appendix C.

**Scene Modeling Module.** We first examine SCA and TCA. Removing SCA causes moderate drops (8.78% SR on seen, 8.53% on unseen), confirming its role in spatial-visual fusion. Without TCA, severe drops occur (19.52% SR on seen, 34.11% on unseen), demonstrating that temporal aggregation is critical for comprehensive scene understanding given the agent's limited camera view. As shown in Table 3, our map semantic segmentation achieves 0.758 mIOU on seen and 0.654 on unseen environments, demonstrating accurate spatial localization of most objects and robustness across novel scenarios. We also test grid resolutions of 80×80, 100×100, and 120×120. 100×100 performs best, while 80×80 and 120×120 cause significant drops (9.64%/8.78% SR on seen, 9.02%/8.41% on unseen). This confirms 100×100 best matches ALFRED's 25m×25m rooms with 25cm steps, where each grid cell corresponds to one agent step. Both components are essential, with TCA most critical.

Table 3: Additional quantitative metrics. mIOU scores for map and image semantic segmentation are presented, while efficiency factors (EF) are compared between SCOUT and w/o Switch Policy.

| mIOU | Seen | Unseen | EF | Seen SR | Seen GC | Unseen SR | Unseen GC |
|---|---|---|---|---|---|---|---|
| Map Semantic Seg. | 0.758 | 0.654 | SCOUT | 0.710 | 0.742 | 0.621 | 0.618 |
| Image Semantic Seg. | 0.869 | 0.779 | w/o Switch Policy | 0.668 | 0.695 | 0.605 | 0.601 |

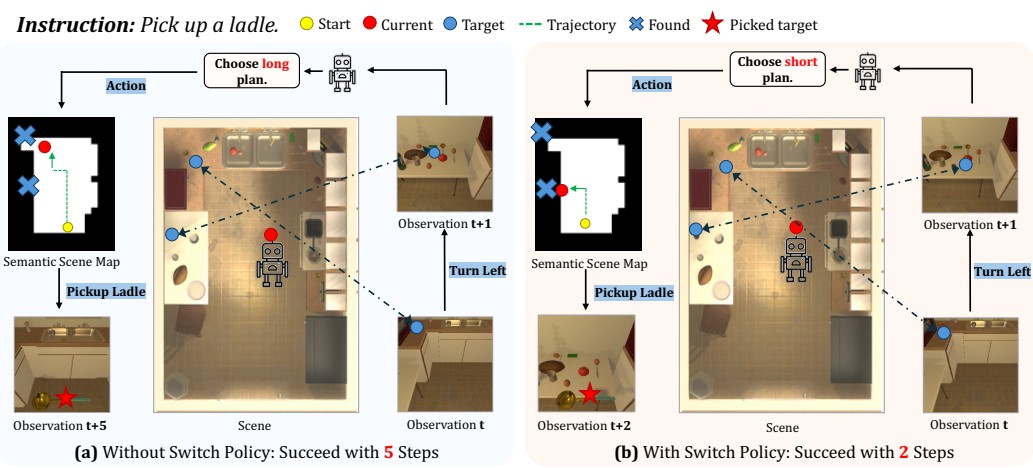

Figure 5: Qualitative comparison of Switch Policy effectiveness. Left: Pickup task completed in 5 steps without Switch Policy. Right: Same task accomplished in 2 steps with Switch Policy.

**Mask Query Module.** Ablation results show the effectiveness of our Mask Query Module in generating precise interaction masks. First, image semantic segmentation achieves 0.869 mIOU on seen and 0.779 on unseen environments (Table 3), showing accurate object localization. Second, comparing with GT masks shows minimal gaps (0.24% SR on seen, 2.55% on unseen), confirming our masks are not the bottleneck for task success. Also, our module uses scene-specific 3D object features, beating static queries (1.34% SR on seen, 2.44% on unseen), which indicates dynamic queries capture object specificity better. Also, failure analysis in Appendix D shows our method reduces interaction failures by 12.6% on seen and 24.3% on unseen tasks versus DISCO(Xu et al., 2024).

**Switch Policy.** Our Switch Policy improves both task success and efficiency. Removing it causes drops: SR falls 4.73% on seen and 4.14% on unseen settings. To measure efficiency, we define the **Efficiency Factor (EF)**: $EF =$ Expert Length/Agent Length $=$ PLWSR/SR. Higher EF shows more efficient navigation with shorter agent paths vs expert trajectories. Table 3 shows consistent EF gains with our Switch Policy across all metrics. Results show Switch Policy gets higher success rates and efficiency by dynamically switching between planning and interaction opportunities.

### 4.4 QUALITATIVE RESULTS

Fig. 5 shows our switch policy effectiveness. Without Switch Policy, agents strictly follow long-term plans, missing optimization opportunities (scenario a). Our switch policy monitors interaction opportunities and interrupts long-term plans when viable interactions are detected, greatly improving SCOUT's efficiency (**5 steps versus 2 steps**). This improvement is confirmed by PLW metrics in Table 1. Further visualization of our scene understanding method is provided in Appendix E.1.

### 5 CONCLUSION

We present SCOUT, an effective framework for embodied mobile manipulation that addresses the challenges of continual scene understanding and adaptive action planning. Through our Spatial-Aware scene representation and Switch Policy mechanism, SCOUT demonstrates superior performance in both seen and unseen environments on ALFRED benchmark. Our ablation studies and thorough visualizations validate the effectiveness of each component. Finally, our method has two limitations: lack of open-vocabulary generalization for objects outside the training vocabulary, and inability to reason about hidden objects contained within other objects.

## 6 ETHICS STATEMENT

We confirm that this research adheres to the ICLR Code of Ethics. We have carefully considered the ethical implications of our work and have strived to conduct our research with the highest standards of scientific integrity and responsibility. We outline the specific considerations below.

1. **Broader Impact and Potential for Harm**
   This research aims to advance embodied AI capabilities for household mobile manipulation tasks. Our work is evaluated exclusively in simulated environments (ALFRED (Shridhar et al., 2020) benchmark), providing a controlled setting for responsible development before real-world deployment.

2. **Data and Privacy**
   Our research utilizes established public dataset named ALFRED (Shridhar et al., 2020), which consists of simulated household environments and synthetic agents interacting with virtual objects. This dataset contains no Personally Identifiable Information (PII) and poses no privacy risks. We commit to releasing our implementation details upon acceptance to promote reproducibility.

3. **Computational Resources**
   Our experiments were conducted on NVIDIA RTX 4090 GPUs. Our SCOUT framework improves efficiency through the Switch Policy mechanism, achieving better task completion rates while requiring fewer steps, thereby reducing computational overhead compared to baseline methods.

## 7 REPRODUCIBILITY STATEMENT

To ensure the reproducibility of our results, we provide comprehensive details of our methodology, experimental setup, and resources. Our core framework, SCOUT, is described in Section 3, with specific architectural details in Section 3.1 and 3.2, and our training strategy also in Section 3. The complete experimental setup, including implementation details, hardware (NVIDIA RTX 4090 GPUs), and key hyperparameters, is detailed in Section 3.1, 3.2 and 4. The datasets used is also described in Section 4. We provide a full breakdown of our evaluation metrics and comparisons against baselines in Section 4.2. To further support our claims, we have included ablation studies in Section 4.3. Furthermore, we commit to releasing our source code upon acceptance of this paper.

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

# A    INTRODUCTION OF ALFRED BENCHMARK

ALFRED (Shridhar et al., 2020) is a large scale dataset which includes 25,726 language directives for embodied mobile manipulation task. Each language directive includes a high-level goal together with low-level instructions.

Given either the goal or instructions in natural language along with egocentric vision, the agent's objective is to produce a series of actions along with object masks which guide interactions with relevant objects in order to complete the task. The agent must fulfill all the conditions required for task completion; even if one condition is not met, the task is deemed unsuccessful.

## A.1    TASK TYPES

All the task in ALFRED can be categorized into 7 types:

1. Look & Examine. Examine an object under the light (e.g. examine an apple under the lamp).
2. Pick & Place. Pick an object and place it in a receptacle (e.g. pick a bowl and place it on the counter).
3. Pick & Place Two. Place two object instances in the same receptacle (e.g. throw two apples into the garbage bin).
4. Stack. Place an object in a movable container then place the movable container in a receptacle (e.g. place a fork in a plate then put the plate on the counter).
5. Heat & Place. Place a heated object in a receptacle (e.g. place a heated egg on the dining table).
6. Cool & Place. Place a cooled object in a receptacle (e.g. place a chilled potato on the dinning table).
7. Clean & Place. Place a cleaned object in a receptacle (e.g. place a cleaned towel in the sinkbasin).

## A.2    SUB-TASKS

All tasks mentioned above can be divided into eight sub-tasks: (1) GotoLocation; (2) PickUp; (3) Put; (4) Slice; (5) Toggle; (6) Heat; (7) Cool; (8) Clean. Each sub-task is a pair of action and target object (e.g. (PickUp, Apple)).

## A.3    ACTION SPACE

The action space of the agent consists of 5 navigation actions (MoveAhead, RotateRight, RotateLeft, LookUp, LookDown), 7 interactive actions (PickUp, Put, Open, Close, ToggleOn, ToggleOff, Slice) and a STOP action indicating the task is completed. Every interactive action necessitates an object mask to identify the specific object for manipulation.

## A.4    EVALUATION METRICS

The evaluation process relies on three key metrics: success rate (SR), goal-condition success rate (GC), and path-length weighted scores (PLWSR and PLWGC). The primary metric, SR, represents the proportion of tasks successfully completed, reflecting the agent's overall task-solving capability. GC, on the other hand, measures the percentage of goal conditions that have been met, providing insight into the agent's ability to achieve partial task success. Lastly, path-length weighted (PLW) scores adjust SR and GC based on the number of actions taken by the agent, assessing its efficiency in task completion.

## A.5    AN EXAMPLE TASK

As illustrated in Fig. 6, this ALFRED example task demonstrates how the dataset provides both high-level directives and step-by-step descriptions for each task. While conventional approaches

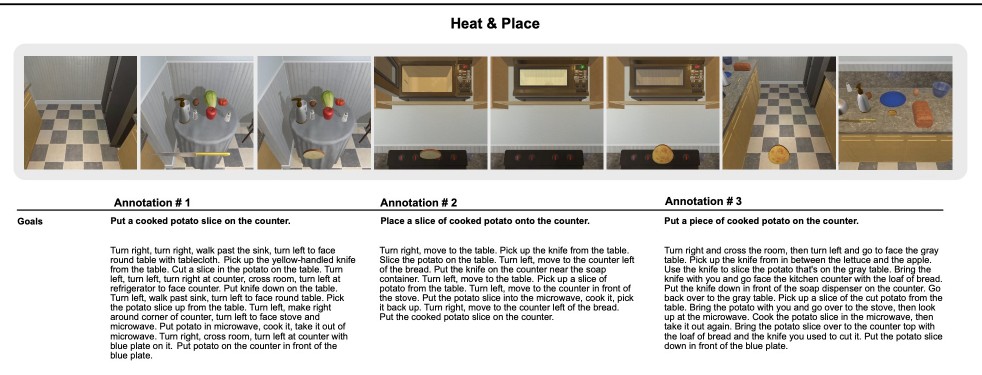

Figure 6: Illustration of an example "Heat and Place" task from the ALFRED (Shridhar et al., 2020) benchmark, showing the agent's sequential interactions with objects in a household environment.

rely on detailed instructions, our SCOUT method efficiently completes tasks using only high-level directives, eliminating the need for granular guidance while maintaining high accuracy and performance.

## B APPLYING SCOUT IN ALFRED

In this section, we describe the comprehensive process of applying SCOUT to ALFRED.

### B.1 LANGUAGE PROCESS

For a fair comparison, we adopt the language processing approach from (Min et al., 2022), which utilizes fine-tuned BERT models to convert natural language instructions into ALFRED's internal parameters. While modern large language models offer potential advantages, our experimental design guarantees an equitable comparison with existing baselines.

First, the BERT model identifies the task type based on the language instruction, then it predicts the following four PDDL parameters: Object Target (the object to be manipulated. e.g. apple), Receptacle (where the object will be finally placed. e.g. table), Movable Receptacle (the intermediate container in stack tasks. e.g. pot), Slice (whether the object needs to be sliced.).

Building on the estimated internal parameters derived from the language module, we leverage ALFRED's structured task framework to construct sub-tasks using predefined templates. The standard templates corresponding to each task type are provided in Table 4. Each sub-tasks consists of a verb-noun pair, which is then processed within the proposed SCOUT framework for execution.

### B.2 AGENT SETUP

The agent's 4-DoF pose is recorded, consisting of a 2-DoF position, a 1-DoF rotation, and a 1-DoF camera tilt angle. At the beginning of each task, the camera is positioned at a 45-degree downward tilt relative to the horizon. As navigation progresses, the agent's pose is incrementally updated based on the accumulation of discrete actions.

At the beginning, the agent scans its panoramic surroundings by performing four consecutive 90-degree rotations. This process allows the agent to build a comprehensive representation of the scene, facilitating the navigation. Once this initial perception phase is complete, the agent proceeds to execute the planned task sub-tasks in sequence.

Table 4: Templates on generating subgoals per task type.

| **(1) Look & Examine** | |
| --- | --- |
| PickUp | Object |
| Toggle | Lamp |
| **(2) Pick & Place** | |
| PickUp | Object |
| Put | Receptacle |
| **(3) Place Two** | |
| PickUp | Object |
| GotoLocation | Object |
| Put | Receptacle |
| PickUp | Object |
| Put | Receptacle |
| **(4) Stack** | |
| PickUp | Object |
| Put | Movable Receptacle |
| PickUp | Movable Receptacle |
| Put | Receptacle |
| **(5) Heat & Place** | |
| PickUp | Object |
| Heat | Microwave |
| Put | Receptacle |
| **(6) Cool & Place** | |
| PickUp | Object |
| Cool | Fridge |
| Put | Receptacle |
| **(7) Clean & Place** | |
| PickUp | Object |
| Clean | Sink Basin |
| Put | Receptacle |

## B.3 Spatial-Aware Continual Scene Understanding

The scene understanding system begins with an egocentric $300 \times 300$ RGB frame, captured by a camera with a 60-degree field of view. To construct a comprehensive scene representation, we leverage a Scene Modeling Module to generate semantic scene map and a Mask Query Module to extract interaction masks. Both map semantic segmentation and image semantic segmentation utilize 87 categories. The Scene Modeling module utilizes spatial cross attention (SCA) to fuse current observations with the existing scene map and temporal cross attention (TCA) to retain information from previous frames. The Mask Query Module produces precise target masks for object interactions by computing pixel-wise similarity between object-specific features and image data. Details of the scene understanding system are in Section 3.1 of the main text.

## B.4 Switch Policy

We employ switch policy when executing sub-tasks, which consists of a long-term planner and a short-term planner. Initially, the agent randomly explores navigable areas until the target object is detected. Upon detection, the long-term planner generates a navigation trajectory to approach the target based on the semantic scene map. Meanwhile, the short-term planner continuously monitors the environment for immediate interaction opportunities. If a closer or more accessible target appears, it interrupts the long-term plan and proceeds with interaction, enhancing efficiency in mobile manipulation. Details of the switch policy are in Section 3.2 of the main text.

Table 5: Failure Analysis.

|  | Valid Seen | Valid Unseen |
|---|---|---|
| Success Rate | 62.32 | 61.27 |
| Language error | 18.29 | 19.24 |
| Object not found | 3.91 | 2.31 |
| Navigation collision | 6.46 | 11.57 |
| Interaction failure | 9.02 | 5.61 |

## C  ABLATION SETTING

**SCOUT.** Our complete model incorporates SCA for spatial-visual fusion, TCA for temporal aggregation, Switch Policy for reactive interaction, 100×100 grid resolution, and dynamically generated object queries from scene features for interaction mask prediction, without using ground-truth semantic segmentation masks from the simulator.

**w/ GT Masks.** This configuration uses ground-truth semantic segmentation masks provided by the simulator for object interaction, while maintaining all other components identical to SCOUT (SCA, TCA, Switch Policy, 100×100 grid).

**w/ SQ.** We replace dynamically computed mask queries (generated real-time from scene features) with fixed static queries - 87 object classes with 256-dimensional vectors per class that remain unchanged after training, while keeping all other components unchanged.

**w/ Grid 80/120.** We test alternative grid resolutions of 80×80 and 120×120 instead of the standard 100×100 configuration, while maintaining all other model components (SCA, TCA, Switch Policy, and dynamic queries) identical to SCOUT.

**w/o SCA.** At time $t$, we directly use the scene feature from time $t-1$ to predict semantic scene map and interaction mask, without SCA updates for spatial-visual fusion, while keeping TCA, Switch Policy, and other components active.

**w/o TCA.** At each timestep, we update scene features based solely on the current observation without incorporating historical scene features for temporal aggregation, while maintaining SCA, Switch Policy, and other components.

**w/o Switch Policy.** We eliminate the Switch Policy mechanism that enables reactive interaction. Without this component, the agent strictly adheres to navigation routes generated by the long-term planner, executing a purely sequential approach to exploration and interaction.

## D  FAILURE ANALYSIS

We conduct a comprehensive failure analysis of SCOUT as reported in Table 5. SCOUT achieves 62.32% success rate on seen environments and 61.27% on unseen environments, demonstrating consistent performance across different scenarios. Language misunderstanding errors constitute the largest proportion of failures, accounting for approximately 18-19% in both settings. This suggests the necessity of more robust language understanding capabilities in embodied AI systems. Object not found failures occur in 3.91% and 2.31% of cases respectively, likely due to target objects being located in closed receptacles or occluded areas that require active exploration with commonsense reasoning. Navigation collision and interaction failure represent other common failure modes, with navigation issues being more prominent in unseen environments (11.57% vs 6.46%), indicating the challenge of generalizing spatial navigation to novel scenes.

## E  MORE VISUALIZATION RESULTS

### E.1  SPATIAL-AWARE CONTINUAL SCENE REPRESENTATION

The visualization results in Fig. 7 illustrates the efficacy of our spatial-aware continual scene representation. In scenario (a), prior approaches fail to detect the CD within the bin, demonstrating their limitations with contained objects. In contrast, scenario (b) demonstrates how our spatial-aware

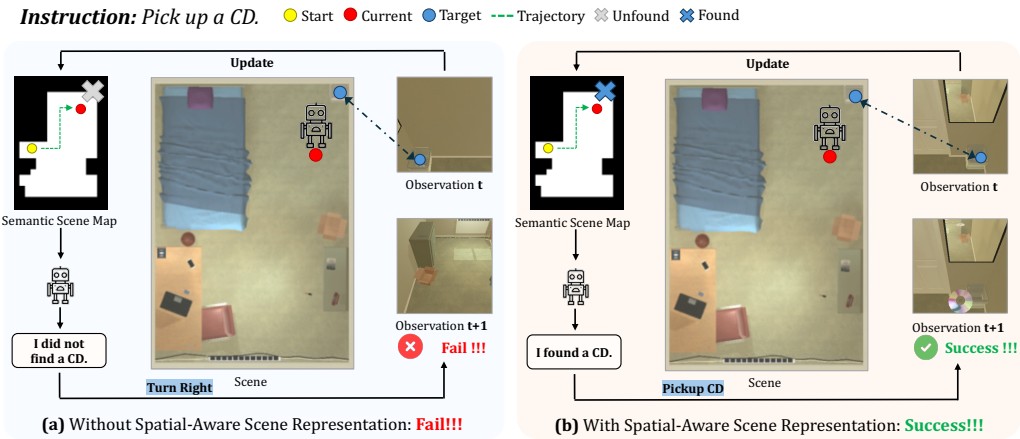

Figure 7: Qualitative analysis of Spatial-Aware Continual Scene Understanding. Left: Failure due to missing spatial map updates. Right: Success with spatial-aware representation.

agent successfully localizes and retrieves the CD from the bin, highlighting the representation's capability in handling complex spatial relationships and containment scenarios.

## F    DECLARATION OF LLM USAGE

In preparing this manuscript, we employed Large Language Models solely for enhancing linguistic quality and textual coherence. All substantive scholarly work, encompassing our research approach, experiment design, and result analysis, represents original contributions from our team. We have thoroughly examined every portion of the text and assume complete accountability for all material presented herein.

