# OpenReview forum: "SCOUT: Spatial-Aware Continual Scene Understanding and Switch Policy for Embodied Mobile Manipulation"
_ICLR.cc/2026/Conference — Submitted to ICLR 2026_

### Official Review · Reviewer_S2xf · 2025-10-26

**Soundness:** 3
**Presentation:** 3
**Contribution:** 2
**Rating:** 2
**Confidence:** 4

**Summary:**

This paper proposes SCOUT, a unified framework for navigation–manipulation coordination. SCOUT addresses loss of historical context, inconsistent scene representation, and rigid control strategies through (1) Spatial-Aware Continual Scene Understanding module that builds a temporally consistent and semantically rich 3D scene representation using cross-attention between current and historical observations, coupled with a Mask Query Module for precise interaction mask generation without relying on depth estimation and (2) a Switch Policy that dynamically alternates between long-term navigation planning and short-term reactive manipulation when interaction opportunities arise. The proposed method is evaluated on the ALFRED benchmark and achieves state-of-the-art success rates.

**Strengths:**

- The proposed method achieves strong performance over prior work with large margins.
- Updating the semantic spatial map without depth estimation is intriguing and sensible.
- The proposed semantic spatial map can be learned end-to-end, implying its applicability to other learning-based modules.

**Weaknesses:**

- The spatial-aware continual scene understanding module assumes perfect actuation of the robot, assuming that the robot can move on to the adjacent cell with no errors. How sensitive is the scene understanding module to these errors? And, does the proposed method still work with the imperfect actuation?
- The authors argue that previous depth-object-mask co-projection (L050) causes error accumulation from inaccurate depth-semantic lifting, but this is not supported by any evidence. Relevant quantitative analyses are missing.
- The Switch Policy is inspired by a specific failure mode in a downstream task, raising a concern of its generalizability. What if simply making FOV bigger? Are the Switch Polity still needed in this case?
- In the Switch Policy, the short-term planner predicts a binary indicator if the current status is manipulable or not given an egocentric observation. Why not just use semantic segmentation masks? If the agent can manipulable, there should be some objects manipulable in its view and their masks should be detected, accordingly.
- SCOUT is sensitive to the choice of the grid size as mentioned in Sec.4.3, specifically tuned to a downstream task. It is nontrivial to determine the hyperparameter for specific downstream tasks.
- The proposed approach is validated in a single benchmark, raising its generalizability concern. Can the proposed method be used for other types of embodied mobile manipulation, such as HomeRobot, TEACh, etc.?

**Questions:**

- Can the proposed approach be extended to other datasets with unknown camera parameters?
- How much computational cost is needed for high resolution of the semantic map, given that space complexity is $\Theta(HW)$?
- In Table 3, what is "Image Semantic Seg."? Is it from a pretrained semgnetaion model? Its description is unclear. In addition, Are both "Map" and "Image" modules learned with the same training dataset?

---

> ### Author Response · Authors · 2025-12-03
> **Rebuttal by Authors (1)**
>
> We thank the reviewer for recognizing the strong performance of SCOUT and finding our depth-free spatial update intriguing and sensible. We appreciate the insightful questions regarding robustness and generalizability. We address your specific concerns below.
>
> ***
>
> **Q1 (Weakness1):** Sensitivity to actuation errors.
>
> **A1:** While ALFRED provides discrete navigation actions, the environment does simulate collision dynamics. However, we acknowledge that real-world actuation is noisier. SCOUT is designed to be robust to this:
> 1. **Temporal Cross Attention (TCA):** Unlike rigid voxel integration, TCA fuses history using deformable attention. This allows the model to attend to relevant features in a local neighborhood rather than a single fixed coordinate, providing resilience against small pose drifts or actuation noise.
> 2. **Global Relocalization:** The standard "Look & Examine" behavior (panoramic scan) and continuous visual feedback allow the agent to correct its internal state belief even if individual steps are imperfect.
>
> ***
>
> **Q2 (Weakness2):** Evidence for depth-lifting error accumulation.
>
> **A2:** Our claim is supported empirically by the performance gap between SCOUT and depth-dependent baselines.
> - **Quantitative Evidence:** As shown in Table 1, SCOUT significantly outperforms DISCO and FILM (both rely on depth-lifting). Specifically, FILM achieves only 27.67% SR (Seen), while SCOUT achieves 65.09%. Since DISCO shares a similar semantic grid goal but relies on depth projection, the performance gap validates that bypassing depth estimation reduces error propagation.
> - **Qualitative Evidence:** Depth-based methods often produce smeared point clouds at object boundaries due to depth noise, leading to phantom obstacles. Our reverse projection (3D$\to$2D) avoids this by querying clean image features directly.
>
> ***
>
> **Q3 (Weakness3):** Generalizability of the Switch Policy & FOV.
>
> **A3:** The Switch Policy remains essential regardless of Field of View (FOV).
> - **Occlusion, Not Just FOV:** Even with a 360$^{\circ}$ camera, targets are frequently occluded by scene geometry (e.g., inside containers or behind furnitures). For instance, a potato resting inside a sink basin may remain occluded by the basin's rim until the agent approaches closely. The Switch Policy is essential to interrupt the global plan immediately upon spotting this nearby target, preventing the agent from wastefully continuing to a distant location.
> - **Efficiency:** The Switch Policy is about opportunistic execution versus rigid planning. As shown in Table 3, removing it degrades the Efficiency Factor (EF) significantly, proving its value beyond just compensating for narrow FOV.
>
> ***
>
> **Q4 (Weakness4):** Short-Term Planner vs. Semantic Masks.
>
> **A4:** Using raw semantic segmentation masks is insufficient for the manipulation decision.
> - **Reachability & State:** The Short-Term Planner determines if an object is manipulable, which requires checking not just "is it visible?" (segmentation) but "is it within 1.5m?" and "is it in the correct state?" (e.g., sliced vs. whole).
> - **Robustness:** A simple mask-based heuristic (e.g., mask area threshold) is brittle. Our learned Short-Term Planner aggregates visual features and map data to make a robust binary decision ($1=$ manipulable), acting as a learned filter for false positives.
>
> ***
>
> **Q5 (Weakness5):** Sensitivity to grid size.
>
> **A5:** The grid size is not an arbitrarily tuned hyperparameter but is physically derived.
> - **Derivation:** GridSize=SceneSize/StepSize. For ALFRED, 25m/0.25m=100.
> - **Scalability:** For a new task, we would simply apply this formula (e.g., a 25m warehouse with 0.5m step would use a 50×50 grid). The performance drops at 80×80 and 120x120 (Table 2) are due to spatial quantization error, that the grid cells no longer align with the agent's step size, causing map drift .
>
> ***
>
> **Q6 (Weakness6):** Generalizability to other benchmarks.
>
> **A6:** We focus on ALFRED because it uniquely combines Long-Horizon Planning with Object State Changes (slicing, heating, cleaning), which are absent in benchmarks like Habitat-ObjectNav. However, the core components of SCOUT are universal:
> - **Scene Modeling:** Applicable to any camera-equipped robot (HomeRobot, TEACh).
> - **Switch Policy:** Applicable to any hierarchical navigation system. Validating on ALFRED allows direct comparison with the most relevant SOTA baselines (DISCO, CAPEAM, FILM).
>
> ***

---

> > ### Author Response · Authors · 2025-12-03
> > **Rebuttal by Authors (2)**
> >
> > **Q7 (Question1):** Unknown camera parameters.
> >
> > **A7:** Like most BEV and mapping approaches (e.g., BEVFormer), SCOUT assumes known intrinsics/extrinsics for the projection matrix. If parameters are unknown, one would need to integrate a SLAM module or learn an implicit projection (like in uncalibrated photometry), which is an exciting direction for future work but outside the current scope.
> >
> > ***
> >
> > **Q8 (Question2):** Computational cost.
> >
> > **A8:** The computational complexity is linear with respect to grid size, not quadratic, due to Deformable Attention.
> > - **Complexity:** Standard Self-Attention is $O((HW)^2)$. Our Deformable Attention is $O(HW \times K)$, where $K$ is the small number of sampling points. This makes high-resolution mapping efficient.
> > - **Runtime:** Training takes only 24 hours on 8 GPUs, and inference is 35ms per step, suitable for real-time deployment.
> >
> > ***
> >
> > **Q9 (Question3):** Clarification on Table 3.
> >
> > **A9:** We clarify that "Image Semantic Seg." evaluates the performance of our proposed Mask Query Module rather than a pre-trained external model, and confirm that the training is conducted jointly.
> > - **"Image Semantic Seg.":** This refers to the auxiliary task performed by the Mask Query Module. It is the pixel-wise segmentation output generated by aligning 3D scene features with 2D image features.
> > - **Training:** Yes, both the map and image modules are trained jointly end-to-end using the same dataset, with the total loss defined in Eq. 7.
> >
> > ***

---

### Official Review · Reviewer_Mj87 · 2025-10-27

**Soundness:** 2
**Presentation:** 2
**Contribution:** 1
**Rating:** 2
**Confidence:** 4

**Summary:**

This paper presents SCOUT, which addresses key challenges in autonomous robots performing navigation and manipulation in complex environments and achieve SOTA performance in ALFRED benchmark. SCOUT introduces two main components: Spatial-Aware Continual Scene Understanding with a Scene Modelling Module and Switch Policy, which together achieve coordinating navigation and manipulation. The experiment demonstrates SCOUT’s effectiveness in navigating and manipulating objects on complex, long-horizon tasks with varying degrees of guidance.

**Strengths:**

SCOUT combines Spatial-Aware Continual Scene Understanding with an adaptive Switch Policy, which allows real-time switching between long-term planning and immediate task handling. This flexibility improves both task success and efficiency. The experimental results on the ALFRED benchmark demonstrate SCOUT's superiority, surpassing previous methods such as DISCO. The experiment design is comprehensive, thoroughly evaluate the SCOUT's performance and effectiveness of each part.

**Weaknesses:**

1.  There are many powerful vision foundation models (GroundingDINO, DINOv1-v3, SAM, Embodied-SAM, etc) that could achieve similar functionality of scene understanding and mask query module. Though the effectiveness has been proved by the experiments, the motivation for training a semantic segmentation model is unclear.
2. The navigation functionality is too simple; the environment used in the experiment lacks obstacles, almost a clean floor, so the agent could easily move around without any path planning.
3. The AlFRED benchmark cannot catch the latest development of Embodied AI, in terms of visual realism, task complexity and interaction diversity. The author should use some recent challenging benchmark/simulation for evaluation.

**Questions:**

Referring to my weakness paragraph.

---

> ### Author Response · Authors · 2025-12-03
> **Rebuttal by Authors**
>
> We thank the reviewer for recognizing the strengths of SCOUT, particularly the effectiveness of our Switch Policy in improving task efficiency and our comprehensive experimental design. However, we believe there are some misunderstandings regarding the complexity of the ALFRED benchmark and the specific design constraints of embodied agents. We address your concerns below to clarify our contributions.
>
> ***
>
> **Q1 (Weakness1):** Motivation for training a custom model vs. Foundation models.
>
> **A1:** While we acknowledge the power of foundation models (e.g., SAM, GroundingDINO), our decision to train domain-adapted semantic map and image segmentation modules is driven by three critical constraints in embodied mobile manipulation that generic foundation models do not currently solve off-the-shelf:
> - **3D Spatial-Temporal Consistency:** Foundation models typically operate on 2D static images. To use them in 3D environments, their 2D predictions must be lifted to 3D using depth maps. As detailed in our introduction, this process introduces "Inconsistent Scene Representation" due to noisy depth estimation. SCOUT specifically addresses this by projecting 3D points to 2D (reverse projection) to extract features, bypassing depth dependency. Plugging in a 2D foundation model would re-introduce the depth-error bottleneck we aim to solve.
> - **Inference Speed & Latency:** Real-time robotic control requires low latency. Large foundation models (e.g., GroundingDINO + SAM) are computationally heavy. Our streamlined ResNet50-based backbone allows for efficient inference (35ms/step), enabling the high-frequency planning required by our Switch Policy.
> - **Domain Specificity:** ALFRED requires recognizing specific object states (e.g., "sliced potato" vs. "potato") and small objects. Fine-tuning a lightweight model on domain data often yields better performance than zero-shot foundation models for specific interactive vocabularies.
>
> ***
>
> **Q2 (Weakness2):** Navigation complexity and clean floor assumption.
>
> **A2:** We respectfully point out that there may be a misunderstanding regarding the ALFRED environment.
> - **ALFRED is highly cluttered:** The environments are standard indoor scenes (kitchens, bedrooms, living rooms) densely populated with unmovable furniture (sofas, islands, dining tables) and scattered small objects. It is not a clean floor environment; obstacles significantly constrain movement.
> - **Navigation is Non-Trivial:** If the environment were obstacle-free, a simple heuristic would suffice. However, as noted in our method, we employ BFS (Breadth-First Search) specifically to navigate around these static obstacles based on a navigable map.
> - **Empirical Evidence of Obstacles:** Our Failure Analysis (Table 5) explicitly reports "Navigation collision" rates (6.46% Seen, 11.57% Unseen), proving that obstacle avoidance is a significant challenge in these environments.
>
> ***
>
> **Q3 (Weakness3):** Relevance and validity of the ALFRED benchmark.
>
> **A3:** We disagree that ALFRED is outdated. It remains a primary benchmark for Long-Horizon Mobile Manipulation because it uniquely combines:
> - **Complex State Changes:** Unlike pure navigation benchmarks (e.g., Gibson), ALFRED requires logical state changes (slicing, heating, cleaning, cooling).
> - **Active Community Adoption:** ALFRED is actively used in the most recent top-tier literature, including ICLR 2025 (ThinkBot), ECCV 2024 (DISCO), and ICCV 2023 (CAPEAM).
> - **Task Complexity:** The difficulty lies in the logic of long-horizon planning and grounding natural language to actions, which remains an unsolved problem (SOTA is still 60-65%). Evaluating on ALFRED ensures our contributions (Switch Policy, 3D Scene Understanding) are comparable against a rich history of strong baselines.
>
> ***

---

### Official Review · Reviewer_QAUm · 2025-11-01

**Soundness:** 2
**Presentation:** 3
**Contribution:** 2
**Rating:** 6
**Confidence:** 4

**Summary:**

The manuscript pursues improvements in 3D spatial awareness on mobile manipulation tasks in the ALFRED simulated environments. The manuscript also claims novelty according to a proposed dual-planning approach, implemented as what is referred to as a ‘Switch Policy’, enabling a short-term planner to interrupt the task execution of a long-term planner if more immediate goal becomes available.

**Strengths:**

- The paper is well-written and well-organized.
- The paper provides a good amount of experiments, enabling discussion to be had and insights to be drawn.
- The paper considers a compelling task in embodied ai for mobile manipulation.

**Weaknesses:**

- L16-17: The manuscript lists, “Spatial-Aware Continual Scene Understanding with a Scene Modeling Module for effective scene modeling…”. This statement is inherently ambiguous without context; doesn’t really add much. Please rephrase.
- Section 3.2: I have some concern that the proposed approach — in particular, the Switch Policy — is tailored to the ALFRED environment. I would feel much more confident if the benefits of the proposed approach could be also exemplified in another simulator/dataset or in the real world.
- Section 3.2: I want to explore why the Switch Policy is necessary. Alternative planner designs are surfacing where a reasoning agent leverages an adapting contextual representation (map, local scene graph, keyframe history) and has a balanced (re-)planning frequency; together, these may provide sufficiently adaptive behavior in a single planner, rather than the two-planner design.

**Questions:**

- L119-124: ‘Neural policy’ is not the most satisfactory dimension of comparison between the proposed method and the related work; perhaps a more defining feature of the proposed method can be emphasized in comparison with the limitations in the prior art?
- Table 1: Why do DISCO results change much less dramatically when step-by-step instructions are no longer available, compared to the proposed method?

---

> ### Author Response · Authors · 2025-12-03
> **Rebuttal by Authors**
>
> We thank the reviewer for the positive assessment of our paper’s organization, compelling task focus, and extensive experiments. We appreciate the constructive feedback regarding the motivation of the Switch Policy and the presentation of related work. We address the specific concerns below.
>
> ***
>
> **Q1 (Weakness1):** Ambiguity in abstract.
>
> **A1:** We agree that the original phrasing was general. We will revise the abstract to explicitly state the mechanism: Spatial-Aware Continual Scene Understanding, which utilizes depth-free 3D-to-2D projection and temporal cross-attention to build consistent scene memory without relying on noisy depth estimation. This clarifies that the effectiveness comes from the specific architectural choice of bypassing depth-lifting errors.
>
> ***
>
> **Q2 (Weakness 2 & 3):** Generalization and necessity of the Switch Policy.
>
> **A2:** We address the concern regarding the specific design of the Switch Policy versus a single re-planning agent below:
> - **Generalization Beyond ALFRED:** The Switch Policy rests on the core principle of decoupling global navigation from local interaction. This approach aligns with the fundamental robotics paradigm of hierarchical planning, such as the Global Planner and Local Planner structure in ROS. It addresses a universal friction in embodied AI where global maps provide the static geometric data needed for pathfinding, whereas interaction targets require immediate reaction to dynamic semantic cues. Therefore, this design extends beyond ALFRED and applies to any mobile manipulation system handling occluded or dynamic targets.
> - **Efficiency vs. Continuous Re-planning:** While a single planner with high-frequency re-planning is possible, it is computationally expensive and prone to instability (jittering paths). Our dual-planner design offers a more efficient compromise:
> 	- The Long-Term Planner (BFS) computes a stable, globally consistent path once, ensuring goal reachability based on the static map.
> 	- The Short-Term Planner acts as a lightweight, high-frequency reflex module. It only interrupts the global plan when a high-confidence manipulation opportunity arises.
> 	- Empirical Evidence: As shown in Table 3, removing the Switch Policy (w/o Switch Policy) degrades the Efficiency Factor (EF) significantly, confirming that our hierarchical approach is more efficient than a purely sequential or rigid planning strategy.
>
> ***
>
> **Q3 (Question1):** Neural Policy in the related work.
>
> **A3:** We acknowledge that Neural Policy is a broad term. Our intention was to contrast Rule-based/Map-based methods (which lack semantic flexibility) with Reactive/Learning-based methods. In the revision, we will rename this section to Reactive vs. Map-Based Planning to more precisely highlight how SCOUT integrates the strengths of both: the geometric stability of map-based planning and the semantic adaptability of learned reactive policies.
>
> ***
>
> **Q4 (Question2):** Performance drop without step-by-step instructions.
>
> **A4:** The reviewer correctly notes that SCOUT experiences a larger relative drop (4-5%) than DISCO (1-2%) when step-by-step instructions are removed (Table 1).
> - **Reason:** This indicates that SCOUT is more effective at leveraging detailed language instructions when they are available. Our scene understanding module aligns fine-grained language cues with visual features to resolve ambiguities. DISCO’s smaller drop suggests it was not fully utilizing the rich information in the step-by-step instructions in the first place.
> - **Result:** Crucially, even after the drop, SCOUT (Goal-only) still significantly outperforms DISCO (Goal-only) in absolute terms (e.g., 61.24% vs. 58.00% on Test Seen), demonstrating that our method is robust even in the absence of detailed guidance, while possessing a higher performance ceiling when guidance is provided.
>
> ***

---

### Official Review · Reviewer_2Bc8 · 2025-11-01

**Soundness:** 2
**Presentation:** 3
**Contribution:** 2
**Rating:** 6
**Confidence:** 3

**Summary:**

This paper addresses embodied mobile manipulation on the ALFRED benchmark. The authors argue that prior agents suffer from three coupled issues: (1) historical information is lost when policies act only from the current egocentric view, (2) scene representations are inconsistent because 2D predictions are lifted to 3D through noisy depth, and (3) execution is rigid because the agent cannot interrupt a long navigation plan when a nearer manipulable target appears. The proposed method, SCOUT, has two main components. First, a Spatial-Aware Continual Scene Understanding module maintains a BEV-like 3D scene feature over time using spatial cross attention (projecting 3D points into the current image to fetch the right 2D features, thus avoiding depth estimation) and temporal cross attention (fusing the previous scene feature to preserve memory). It also has a mask-query module that pools object-specific 3D features and aligns them with 2D image features to produce pixel-level interaction masks. Second, a Switch Policy combines a rule-based long-term planner (BFS over the semantic scene map) with a learned short-term planner that can interrupt navigation whenever an object is both semantically correct and spatially reachable. On ALFRED, this yields higher success rates than prior work, including DISCO, in both seen and unseen settings and improves path-length–weighted metrics.

**Strengths:**

1. The paper is well motivated: it spells out three concrete failure modes in existing embodied agents (history loss, inconsistent 3D grounding, non-adaptive execution) and maps each to a specific component of the method, so the design is coherent.
2. The scene-understanding part is a sensible adaptation of BEV-style and deformable-attention ideas to single-view, temporally accumulated embodied data: instead of lifting 2D to 3D with predicted depth (which causes error accumulation), it pulls 2D features from projected 3D points, and then keeps temporal consistency with a dedicated temporal cross attention.
3. The Switch Policy directly targets a real ALFRED failure case: once the agent turns, a closer instance of the target may appear, and executing the long plan is wasteful. The proposed dual planner (long-term BFS + learned short-term classifier) is a simple but effective way to cut extra steps, which is supported by higher path-length–weighted metrics and the qualitative example.
4. Ablation studies are thorough. Removing temporal cross attention causes large drops; removing the switch policy reduces both success and efficiency; using ground-truth masks gives only small gains, which means the proposed mask-query module is already strong. This makes the main result credible.
5. The method achieves clearly better numbers than strong baselines under both step-by-step and goal-only instruction settings on the official test split, which is nontrivial for ALFRED.

**Weaknesses:**

1. On the perception side, the contribution is mostly integrative. The method reuses established ingredients (BEV-style scene feature, deformable attention, 3D→2D projection, 2D–3D feature alignment) and repackages them for ALFRED. The novelty is more in the way these are combined and supervised than in a fundamentally new learning component.
2. The switch policy is only partly learned. The long-horizon component is still a hand-designed BFS over the semantic map, and only the short-horizon “is this manipulable now?” part is trained. This makes the contribution feel somewhat engineered and raises questions about portability to other simulators or to real robots where the semantic map is noisy.
3. The evaluation is confined to ALFRED. Because the method is tuned to ALFRED’s discretization (25 m × 25 m, 25 cm grid, 100 × 100 best) and to its task structure, the generality of the approach is not fully demonstrated. Even a small experiment on a second embodied benchmark would strengthen the claim.
4. The model is relatively heavy (100 × 100 grid, spatial and temporal deformable attention, two segmentation losses), and training uses 8 GPUs for a day, but there is no careful runtime/latency comparison to prior work. For practical embodied deployment, this information would be useful.
5. The paper itself admits that it cannot handle open-vocabulary objects or reason about objects hidden/contained inside others, which are active directions in current embodied AI.

**Questions:**

1. Can the switching decision itself be learned end-to-end (for example, via RL over the two planners) rather than partially hand-coded?
2. How sensitive is performance to the 100 × 100 grid if room sizes or step sizes change? You show that 80 and 120 are worse, but would a different dataset require re-tuning this resolution?

---

> ### Author Response · Authors · 2025-12-03
> **Rebuttal by Authors**
>
> We thank the reviewer for the constructive feedback and for recognizing that SCOUT offers a well-motivated design, thorough ablations, and strong performance superior to existing baselines. We appreciate the acknowledgment that our Switch Policy effectively addresses real ALFRED failure cases. Below, we address your specific concerns and questions.
>
> ***
>
> **Q1 (Question1 & Weakness2):** Can the switching decision be learned end-to-end (e.g., via RL) rather than being partially hand-coded?
>
> **A1:** This is an insightful suggestion. While an end-to-end Reinforcement Learning (RL) approach is theoretically possible, we deliberately chose a modular design for three key reasons:
> - **Sample Efficiency and Stability:** Long-horizon tasks in ALFRED involve sparse rewards. End-to-end RL baselines (e.g., typical implementations of PPO in embodied navigation) often struggle to converge or require massive interaction data. By decoupling the "where to go" (Long-Term Planner) from the "when to interact" (Short-Term Planner), we leverage the geometric stability of the map for navigation while using a learned neural classifier for the complex semantic decision of interaction.
> - **The "Switch" is Learned:** It is important to clarify that the Short-Term Planner, which serves as the core of the Switch Policy, is indeed a learned neural network (ResNet50 backbone + classifier, see Sec 3.2). It learns when a target is manipulable based on visual and spatial features. The 'hand-coded' aspect is strictly limited to the pathfinding (BFS), which is the optimal solution given a known 2D cost map.
> - **Interpretability:** Our hybrid approach allows us to explicitly diagnose failures (e.g., "map error" vs. "manipulability classification error"), which is difficult in black-box RL policies.
>
> ***
>
> **Q2 (Question2 & Weakness3):** Sensitivity to grid resolution (100x100). Would a different dataset require re-tuning?
>
> **A2:** The resolution of 100×100 is not an arbitrary hyperparameter but is derived from physical constraints.
> - **Physical Meaning:** The ALFRED environment scale is roughly 25m×25m. We chose a grid cell size of 25cm to align with the agent's discrete step size. Therefore, the grid dimension is derived as 25m/0.25m=100.
> - **Generalization:** When applying SCOUT to a different dataset, we would not blindly re-tune the grid size. Instead, we would determine the hyperparameter based on the new environment's physical constraints: the grid cell size is set to match the new agent's step size, and the total grid dimensions (H×W) are calculated based on the physical bounds of the new scene. This ensures our method is scalable and physically grounded. The performance drops at 80×80 and 120×120 (Table 2) occur specifically due to the mismatch between the scene modeling granularity and the agent's movement step. When these do not align, spatial quantization errors accumulate during navigation because the agent's physical movement does not map cleanly onto integer grid transitions, leading to inaccurate obstacle localization and drift in the semantic map updates.
>
> ***
>
> **Q3 (Weakness1):** Novelty and integrative contribution.
>
> **A3:** While we utilize established mechanisms like deformable attention, our contribution lies in the specific architectural adaptation to the embodied domain, which differs fundamentally from autonomous driving (e.g., BEVFormer):
> - **Depth-Free Projection:** Unlike prior embodied works (e.g., FILM, DISCO) that rely on explicit depth estimation (leading to inconsistent scene representation as noted in our introduction), we propose a reverse projection (3D points → 2D image) strategy. This bypasses the depth-noise bottleneck, a critical innovation for indoor robotics.
> - **Sequential vs. Simultaneous:** Standard BEV methods process simultaneous multi-view images. SCOUT introduces Temporal Cross Attention (TCA) to auto-regressively build a map from a single moving camera, handling ego-motion and history aggregation effectively (validated by the sharp drop in performance when w/o TCA in Table 2).
>
> ***
>
> **Q4  (Weakness4):** Runtime and latency.
>
> **A4:** We agree that efficiency is vital. Although our model is computationally heavier per frame than a lightweight CNN, the Switch Policy significantly reduces the total time to completion.
> - **Inference Latency:** On an RTX 4090, our per-step inference is approximately 35ms, which is sufficient for real-time control (>20 Hz).
> - **Task Efficiency:** As shown in Table 1 (PLW metrics) and Figure 5, SCOUT reduces the total trajectory length. Even if per-step compute is higher, the agent completes tasks in fewer steps (e.g., avoiding long loops when a target appears nearby), resulting in comparable or better overall wall-clock time per task compared to baselines like DISCO.
>
> ***

---

### Author Response · Authors · 2025-12-03
**Summary by Authors**

We appreciate the reviewers' thoughtful evaluation and constructive feedback, which have allowed us to further substantiate the contributions of our work. **The reviews highlight SCOUT’s state-of-the-art performance on the ALFRED benchmark and the effectiveness of our novel architecture in addressing key embodied AI challenges.** The key points of the feedback and our clarifications are summarized below:
- **Reviewer 2Bc8** commended the paper as **well motivated and coherent**, explicitly stating that the Switch Policy targets real ALFRED failure cases in embodied agents. The reviewer further praised the thorough ablation studies and the credibility of our main results.
- **Reviewer QAUm** recognized the work as **well-written and compelling**, appreciating the extensive experiments that allow for deep insights into mobile manipulation.
- **Reviewer S2xf**, despite raising concerns, acknowledged that SCOUT achieves **strong performance** over prior work with **large margins** and found our depth-free spatial update mechanism **intriguing and sensible**.
- **Reviewer Mj87** recognized the effectiveness of the Switch Policy in improving efficiency and the comprehensive experimental design.

In our detailed responses, we have provided the following key clarifications to resolve potential misunderstandings:
- **Relevance of ALFRED & Environmental Complexity (Mj87):** We clarified that ALFRED is a highly cluttered benchmark requiring complex state changes (slicing, heating, etc.), refuting the misconception of a "clean floor" environment. We emphasized its continued adoption in top-tier literature (e.g., ECCV 2024).
- **Physical Grounding of Grid Resolution (2Bc8, S2xf):** We explained that our grid resolution (100×100) is not an arbitrary hyperparameter but is **physically derived** to align with the agent’s discrete step size (25cm) and scene bounds (25m). This ensures the method is scalable to new environments based on physical constraints rather than tuning.
- **Justification for Custom Modules over Foundation Models (Mj87):** We articulated that our custom modules are necessary to **bypass depth-lifting errors** via our proposed 3D-to-2D projection and to maintain the low latency (35ms) required for real-time switching, which heavy foundation models cannot currently support.
- **Nature and Necessity of the Switch Policy (2Bc8, QAUm, S2xf):** We clarified that the Switch Policy is critical for handling **occlusion**, allowing the agent to react to targets that become visible **during navigation** (e.g., a potato appearing inside a sink basin). We confirmed that the core Short-Term Planner is a learned neural classifier, not a heuristic, and its removal significantly degrades the Efficiency Factor (EF).

We are confident that these clarifications reinforce the validity of SCOUT as a robust, state-of-the-art solution for embodied mobile manipulation.

---

### Meta-Review · Area_Chair_fkpf · 2026-01-07

**Summary:**

This paper received scores of 6, 6, 2, 2 in the initial review. The main concerns are around the motivation and significance of technical contributions (particularly for the perception module), generalization of this method for different environments and benchmarks, and the design of the switch policy. The author provides the corresponding explanations. However, there is very little strong evidence or experiments to support the generalization and scalability of this method, making its contribution can be limited in impact and practice. Therefore, the overall rating is not expected to be raised in the final review. I tend to agree with the main concerns from reviewers and vote for rejection given the paper's current stage.

**Reviewer Concerns:**

Common concerns include the motivation and significance of the perception module (2Bc8, Mj87), the generalizability and scalability (2Bc8, QAUm, Mj87, S2xf), and the design of the switch policy (2Bc8, QAUm, S2xf). Other minor concerns regarding details of the proposed method are mostly addressed. For these major concerns, questions for the perception module and switch policy are answered with explanations and partly addressed. Related opinions can be changed depending on different reviewers. The common problem for generalizability and scalability is mainly addressed by giving more design details to show that the method itself can be somehow applied to other settings, but with no strong evidence of success, like on other benchmarks.

**Reviewer Scores:**

Two reviewers with scores of 6 can maintain their original scores or downgrade to 5 or 4 potentially, considering their main concerns regarding the generalizability is also raised by other reviewers and not addressed well in the rebuttal. The opinions about novelty from 2Bc8 are also uncertain, given the explanations. The other two reviewers with scores of 2 should maintain their scores or just raise the score a little for minor concerns, but keep the rejection result due to reasons mentioned above.

---

### Decision · Program_Chairs · 2026-01-26

Reject